

# How to get your message through? Designing an Impactful Knowledge transfer plan in a European Project

Sara Pasqualetto[1], Luisa Cristini[2], and Thomas Jung[3]

[1]Alfred Wegener Institute Helmholtz Centre for Polar and Marine Research, Germany

**Correspondence:** Sara Pasqualetto (sara.pasqualetto@awi.de), Luisa Cristini (luisa.cristini@awi.de)

**Abstract.** Academic research is largely characterized by scientific projects striving to advance the understanding in their respective fields. Financial support is often subjected to the fulfilment of certain requirements, such as a fully developed knowledge transfer plan and dissemination strategy. But the evaluation of these activities and their impact is rarely an easy path to clarity and comprehensiveness, considering the different expectations from project officers and funding agencies, dissemination activities and objectives, and so on. With this paper, based on the experience of the management and outreach team of the EU-H2020 APPLICATE project, we aim to shed light on the challenging journey towards impact assessment of knowledge transfer activities by presenting a methodology for impact planning and monitoring in the context of a collaborative and international research project. Through quantitative and qualitative evaluations and indicators developed in four years of the project, this paper represents an attempt to build a common practice for project managers and coordinators and establish a baseline for the development of a shared strategy.

## 1 Introduction

Assessing the impact of public-paid initiatives and projects is very important to gather a better understanding of, among other things, the relevance of the initiative and its subject matter, the success in enhancing knowledge on a particular issue and the effectiveness of the use of funding. Especially in the context of scientific research, measuring impact has become an increasingly relevant aspect for funding agencies, in the conceptualisation of proposals as well as in the reporting phase. When evaluating the impact of research initiatives, for example on policy-making and industrial practices, there are many aspects to consider, such as the purposes of the assessment, the contexts in which it is needed, as well as its scale (Morton, 2015).

Literature in this regard exists, with detailed and insightful examples coming especially from the non-peer-reviewed literature of research projects and funding institutions. For example, a 2015 document from the European Commission dealing with Horizon 2020 indicators defines impact as "the wider societal, economic or environmental cumulative changes over a longer period of time," for which "impact indicators represent what the successful outcome should be in terms of impact





on the economy/society beyond those directly affected by the intervention" (European Commission, 2015). When discussing
methodologies to assess the impacts of higher education institutions on sustainable development, Findler et al. (2019) state that
"impacts refer to the effects that any organization, such as a [Higher Education Institution], has outside its organizational or
academic boundaries—on its stakeholders, the natural environment, the economy, and society in general." Moreover, impact
can concretise in different areas, from economy to policy-making; and it can be directly or indirectly attributed to a project or
institution (Findler et al., 2019). But why is it important to keep track of a research project's impact and provide measurable
assessments to evaluate it? For Upton et al. (2014), impact assessment serves a double function: it provides proof of success of
an initiative, project or institution; and it works also to incentivize future endeavours towards enhancing this impact.

There are many examples that present different strategies and approaches to assessing the impact of research initiatives
or institutions. However, a generalised assessment on how to tackle this key aspect when applied to the evaluation of the
performance of knowledge transfer (KT) activities is still lacking. We refer here to the definition provided by the University of
Cambridge (2009), which defines KT as "the transfer of tangible and intellectual property, expertise, learning and skills between
academia and the non-academic community." Some contributions dealing with measuring the impact of KT efforts have looked
for example into the impact of single dissemination practices, such as the use of posters, on the success of knowledge transfer
(Rowe and Ilic, 2009), or have discussed the evaluation of knowledge transfer activities and their impact on larger policies
involving the agricultural sector and public funding (Hill et al., 2017), but there is no comprehensive attempt to present a
fully developed methodology for the evaluation of KT actions in research. Especially in the panorama of research funding
and science projects, guidelines on the matter are often unclear and largely based on the application of a set of quantitative
metrics known as Key Performance Indicators (KPI), which often provide insightful knowledge on the reach of each activity,
e.g. the number of people who visualise a post or attend a talk, but are not always adequate to measure the uptake of this
activity, meaning for example the number of people that make use of that knowledge ((Morton, 2015)). Moreover, different
requirement to assess impact in KT initiatives might be subjective to the project advisers overseeing the performance of the
individual projects and therefore differ among advisers even within the same funding agency. Finally, the impact assessment
of KT aspects are not playing a defining role in the conceptualisation of communication and dissemination strategies, as they
are usually relegated to being a collection of numerical indications during the reporting phase.

With this paper, we aim to address this issue and present a road map for the development of a successful impact plan for
research projects. We will outline our methodology as it was implemented in the APPLICATE project, present the outcomes
and the strengths of this method and discuss some lessons learned from the project.

To do so, in the following sections we will describe the APPLICATE project, its purpose and its messages, to outline the
context in which this work came about. An overview of APPLICATE's knowledge transfer (KT) strategy will follow as applied
in four different aspects: communication and dissemination, user and stakeholder engagement, training and clustering. In the
fourth section of the paper we will present the project's methodology to monitor and report impact in a quantitative and
qualitative manner, and close with key lessons learned and recommendations, and some conclusive remarks.





## 2 APPLICATE Key Messages

APPLICATE is a European project funded through the Horizon 2020 Research Programme, involving fifteen research institutions, universities and national weather centres from eight European countries and Russia. The project, which ran over 4.5
years and ended in April 2021, has brought together expertise from scientific communities working with weather and climate models to assess the impact of Arctic changes in mid-latitudes, and to respond to specific stakeholder needs for enhanced predictive capacity in the Arctic region from weather to climate time scales. A wide range of stakeholders and users were therefore included in the project through the User Group, and other project outreach activities. They were regularly consulted to effectively exchange knowledge on the latest science and user needs.
Throughout the project lifetime, the APPLICATE community has

- developed and advanced cutting-edge numerical models and produced climate assessments which contribute to the Intergovernmental Panel on Climate Change (IPCC) Assessment Report 6 (AR6). APPLICATE has contributed to the assessment of weather and climate models (including models participating in the Coupled Model Intercomparison Project Phase 6, CMIP6) in the Arctic (Notz and Community, 2020). This along with the availability of new freely available
software will result in critical recommendations for future model development efforts, leading to better forecasts and projections (Blockley and Peterson, 2018).

- delivered novel datasets contributing to the World Meteorological Organisation's Year of Polar Prediction (YOPP) (Bauer et al., 2020) to assess the impact of Arctic climate change on the rest of the Northern Hemisphere (Polar Amplification Model Intercomparison Project, PAMIP) (Smith et al., 2019) and provided a consolidated view of the underlying mech-
anisms and sources of uncertainty.

- provided detailed recommendations for an optimized Arctic observing system taking into account different needs such as forecasting and monitoring (Ponsoni et al., 2020; Keen et al., 2021; Lawrence et al., 2019).

## 3 Knowledge transfer activities

In APPLICATE, Knowledge transfer (KT) is intended as the process through which the knowledge and results produced within
the project are shared with relevant groups and individuals within and outside the project to help them address the challenges of climate change while also fostering innovation, economic and societal growth. Knowledge transfer within APPLICATE focused on four main (groups of) activities: (1) outreach, communication and dissemination, (2) stakeholder engagement, (3) training, and (4) clustering.

Specific objectives of knowledge transfer were to

1. increase public awareness about the impact of Arctic changes on the weather and climate of the Northern Hemisphere (Outreach);





2. develop relevant forms of communication within and outside the EU to adequately convey the results and recommendations to inform policy and socioeconomic actions (Communication);

3. maximise exposure of the science produced to end-users, stakeholders and the public at large, and communicate project results, in order to assure knowledge sharing and knowledge exchange with stakeholders (Dissemination);

4. contribute to servicing those socioeconomic sectors in the Northern Hemisphere that benefit from improved forecasting capacity (e.g. shipping and energy generation) at a range of time scales, as well as enhancing their capacity to adapt to long-term climate change (Stakeholder Engagement);

5. improve the professional skills and competences for those working and being trained to work within this subject area (Training).

6. ensure effective collaboration with partners from Europe and the wider international community (Clustering)

In the following, we summarise the main points of APPLICATE outreach, communication and dissemination plan, user engagement plan, training plan, and clustering plan. For each plan we describe:

– The goal of the KT activity

– The target audience

– The team or person in charge of coordinating the KT activity

### 3.1 Outreach, communication and dissemination

Outreach, communication and dissemination activities of the APPLICATE project were carried out within the Work Package 7, although each work package was required to contribute to the efforts of disseminating results and engage in communication activities. As APPLICATE scientists strived to understand the linkages between changes in the Arctic climate and weather and climate in the mid-latitudes, the communication team of the project sought to increase awareness about these linkages as well as the policy-relevance of the research, and aimed to maximize the impact of scientific findings to end-users. Moreover, through targeted activities directed at different users and stakeholder groups, APPLICATE communication and dissemination sought to establish and maintain an effective dialogue with a network of key stakeholders for a mutual exchange of information on the project and feedback on its activities. Thus, communication activities within the project were essential not only to produce an "outbound" of information (from the project scientists to the broader community), but also for an "inbound," to collect feedback and valuable impressions from the users of weather and climate services as well as stakeholders in the Arctic region and beyond.

In order to fulfil these objectives, the communication team identified key tools and platforms (some of which shortly explained below) to strengthen the project results while keeping open the information flow from the public. Each of the activities was dedicated to convey specific messages and information to determined target groups.





The first, most prominent channel of communication was the project website[1]. It served as a "entry-level user interface" (Hewitt et al., 2017), central server for materials, information, activities carried out in the framework of APPLICATE. In the information infrastructure of the project, the website represents the central hub: all the side activities pass through the website, which functions as re-directory platform for more specific communication activities. The project website is designed to attract all audiences each of which will then navigate it in different ways to find the information relevant for them.

Publications are the primary channel for scientific findings and thus key instruments for the broad dissemination of the project and its relevance particularly among the scientific community. They are indicators of the scientific excellence of the project and provide the "manifesto" of the APPLICATE research for other scientists and projects.

Another fundamental tool for the dissemination of project results and research were conferences and meetings. These occasions represented opportunities that were essential for the presentation of the work carried out in APPLICATE to various audiences and groups, from fellow scientists, to industry and policymakers. As Hewitt et al. (2017) describe in their paper, "such activities enabl[e] co-learning and co-development of products and services." This is an aspect of the communication strategy that heavily relies on the involvement of scientists and researchers, to present their studies and connect with the community.

The project has also established its presence on social media: the APPLICATE Twitter profile has been set up with the objective to communicate in a quick and relatable way the events, activities and progress of the project, to relate with a broader public that goes beyond the academic, political or industrial frameworks, and ultimately to maintain an easier contact with projects and institutes that carry out similar studies.

Other information materials have then been produced to convey specific information to targeted audiences, often in institutional frameworks. These documents include e.g. policy briefs, which summarize valuable information presented in a way that would especially target policymakers and governmental officials.

At the beginning of the project a list of target stakeholders addressed by the APPLICATE project and their communication requirements has been developed which included the target audience of the communication, objective and content of the communication, the type of language (e.g., technical, functional, industry-dependent, non-specialist), the primary mean of the communication along with the timing and the responsible team member to guide all the team members and especially the Executive Board, the coordination team and the communication and dissemination team (Johannsson et al., 2019).

## 3.2 Stakeholder engagement

The stakeholder and user engagement activities of the APPLICATE project were carried out within the Work Package 7 too and described in the Deliverable 7.3 User Engagement Plan (Bojovic et al., 2019). The terms "stakeholder" and "user" have been used interchangeably and include all those that can be interested in and/or can benefit from better knowledge of the weather and climate in the Arctic and the Northern Hemisphere. By pro-active user-engagement, the latest advances in forecasting system development can be effectively communicated to and benefit those economic sectors and social aspects that rely on improved forecasting capacity.

---

[1]https://applicate-h2020.eu/



To develop and conduct targeted user engagement activities and foster co-production of knowledge in the project, we have
divided users in three categories:

1. Key users – Business and governmental stakeholders in the Arctic, within and outside the EU

2. Primary users – Scientific community, meteorological and climate national services, NGOs and local and indigenous communities

3. Secondary users – Business stakeholders from mid-latitudes.

The APPLICATE community increased the stakeholder-relevance of its research by applying a co-production approach (Bojovic et al., 2021) that continuously took into account user needs and feedback to the APPLICATE results via the User Group, workshops, meetings, interviews with key stakeholders, virtual consultations, and development of case studies and policy briefs, hence directly impact by improving stakeholders' capacity to adapt to climate change.

A group of stakeholder representatives and users of climate information (User Group) has been set up at the beginning of the
project, including Arctic stake- and rightholders (e.g. businessess, research organisations, local and indigenous communities). The User Group acted as an external advisory board external to the project that through the stakeholder engagement team could counsel the scientists on their information needs, existing gaps in data, and more widely the issues they encountered in adapting to climate change and where the project could provide results. User Group members also learned directly from the project the latest information available on Arctic climate change, thereby serving as "ambassadors" of the project team to their
respective communities of practice.

Stakeholder engagement was primarily led by the stakeholder engagement team within WP7 who worked at the interface between users of climate data information and the scientists within the project. This link worked in both directions: from the project to the users to inform on the scientific results and their applicability in various societal sectors, and from the users to the scientists within the project to collect information on the needs and requirements, identify research gaps and work together
to co-produce relevant scientific outcomes.

Early in the project, a list of key stakeholders with whom to interact and to target with tailored communication and dissemination activities had been identified. From EU policy makers, to providers of climate services, industry and civil society, the work package leaders have identified possible audience-groups that have interests in the research carried out within APPLICATE. Not only the target groups would be end-users of products and results developed by the scientists working on the
project, but they would also provide feedback on the work done and help design a path for future research and developments.

An example of activity to engage with users of climate information is the blog "Polar Prediction Matters"[2] initiated by APPLICATE together with WMO initiative Year of Polar Prediction (YOPP) and the H2020 project Blue-Action, hosted on the Helmholtz Association website. This platform has been built with the aims to reach out to end-users of weather and climate services in the polar regions and facilitate a dialogue between those who research, develop and provide polar environmental

---

[2]https://blogs.helmholtz.de/polarpredictionmatters/




forecasts and those who (could) use these results to guide their decisions. It has been a key tool to gain feedback on the products and services developed by the projects and forecast centres.

### 3.3 Training

The training activities of the APPLICATE project were carried out within the Work Package 7, too. Training not only aimed to improve the professional skills and competences of those working and being trained to work within the project, but it also
provided a legacy for future generations of scientists and experts working in the fields of climate and weather prediction and modelling.

As part of WP7, the Association of Polar Early Career Scientists (APECS) in cooperation with APPLICATE scientists and coordination team developed and conducted comprehensive and targeted in-person and online training events and resources for early-career researchers, addressing a variety of topics and skills, some of which specific to the subject of polar prediction,
some other transferable (e.g., project management, science communication).

To optimise efforts, a training plan had been developed at the beginning of the project with a list of possible training activities along with their timing and modalities of delivery which was updated throughout the project's lifetime (Fugmann et al., 2019).

Examples of training activities developed and delivered within the APPLICATE project include online seminars, training sessions at the project General Assemblies, a summer school, training workshops and an online course. All activities were
followed by feedback sessions or surveys and the results were collected in deliverable 7.10 "Assessment of early career researchers training activities" (Schneider and Fugmann, 2020).

### 3.4 Clustering

The clustering activities of the APPLICATE project were carried out within the Work Package 8 to facilitate coordination and exploit synergies for a number of European and international activities that are related to some of the activities planned in
APPLICATE. The strategy employed was to focus on a limited number of clustering activities that were given full attention with the aim to ensure effective collaboration with partners from Europe, North America and the wider international community.

A Clustering Plan has been developed at the beginning of the project to identify major European and international projects and players focused at advancing polar prediction capacity and understanding the impact of climate change in the Arctic (Jung et al., 2019). In the plan, updated throughout the project's lifetime, each target cooperating project and initiative is linked with
a project's team member who is responsible for the collaboration and to report to the Executive Board on clustering activities.

Clustering within APPLICATE has been done on both the scientific and the coordination levels. At the scientific level, experiments and publications have been developed together with the international Arctic modelling community (e.g., with the "EU modelling cluster" of projects funded by the European Commission, or with projects participating in the CMIP6-endorsed Polar Amplification Model Intercomparison Project and in the Sea Ice Model Intercomparison Project) and involved primarily
the project scientists and task leaders. At the coordination level, clustering was established through organisation of events, preparation of documents and activities with other projects and programs focused on the polar regions but not necessarily on numerical modelling (e.g., the "EU Polar Cluster" of projects funded by the European Commission or activities in cooperation





with the WMO's Polar Prediction Project) and involve primarily the coordination team, the stakeholder engagement team and the communication team.

## 4   Impact Tracking

The main purposes of KT activities in APPLICATE include maximising exposure, increasing awareness and conveying key results to the wider community of researchers, policymakers and businesses. Therefore, we consider "impact" any significant advancements in expanding the community awareness related to Arctic–mid-latitude linkages and Arctic weather and climate research, disseminating the results of APPLICATE's research and, in general, enlarging the horizons for knowledge exchange and research uptake. In this section, we list the aspects of the project's KT strategy that were instrumental towards this objective, and we present, for each of them, a set of quantitative and qualitative indicators to assess the impact of the action.

### 4.1   Outreach and dissemination

In the APPLICATE project, we made use of different tools to measure and assess the impact of our communication activities. Different instruments may be applied to different types of initiatives or action, and here we illustrate how these have been applied in APPLICATE.

#### 4.1.1   Dissemination and Outreach activities

Meetings, conferences and workshops, press releases, webinars and videos are all fundamental platforms for the dissemination of project results. The impact of these activities can not only be measured in terms of how many events have been attended or organised by project scientists, but it is also worth considering the nature of the event (i.e. the type of event, size, targeted community, type of arrangement, etc.): for example, an international conference has a higher impact index than a project meeting, considering both the number of people and the variety of audiences and disciplines they may attract. To gather the most information on the reach and uptake of these events, it is recommended to monitor and evaluate meetings and workshops not only after their conclusion, but also during the event itself and before (Hewitt et al., 2017).

The project has therefore set up a database to collect information about the event or initiative organized by participants of the project or to which project partners participated and presented the project results. The information requested and collected in the spreadsheet are based on criteria required by the European Commission in its project portal and are graphically summarized in Fig. 1. For each entry, the database shows key identification information, such as

1. the title of the meeting or initiative, together with the date and place of occurrence;

2. the type of action, which can entail the participation to or organization of a conference, workshop, or event other than these; non-scientific and non-peer reviewed publications, press releases; flyers, exhibitions, videos or films; social media campaigns and other media presence, e.g radio or TV interviews; participation to or organization of brokerage and pitch events;





3. the primary and secondary audience attending the event;

4. an estimated number of people reached during the initiative;

245   5. type of APPLICATE users targeted by the initiative.

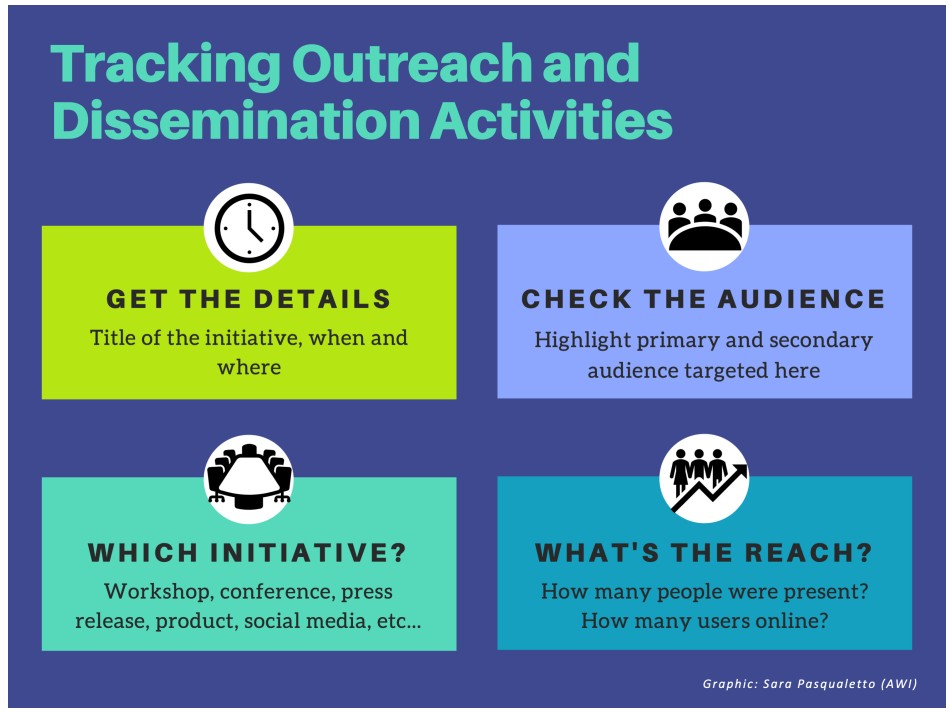

**Figure 1.** Schematic summary of the most relevant aspects to track and evaluate the performance of outreach and dissemination activities.

The main objective of this database is to have a comprehensive overview of the places and audiences reached by APPLICATE initiatives to be able to assess the scope of these efforts on the one hand and to evaluate the pertinence with the communication strategy and its objectives on the other. The latest point would include an evaluation of the target audience, i.e. if the events organized or attended reached out to the intended target group, and if there are groups that these initiative fail to engage, 250   therefore opening the path towards a correction of the strategy and the development of an ameliorated plan.

### 4.1.2   Zenodo and Google Scholar: the publications archives

In academic research, peer-reviewed publications are fundamental indicators of scientific impact. The number of publications released in the framework of the project as well as their reach (How many citations did they generate? Which journals have we published in?) make up a significant part of the puzzle when discussing communication impact. 255   The APPLICATE project collects all its relevant published and unpublished work in the online repository Zenodo. This online platform developed by CERN and OpenAIRE provides a community space where publications, conference proceedings

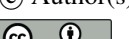


such as abstracts, presentations and posters as well as project documents like Deliverables are streamed into a collective space. All participants to the APPLICATE project may upload directly their contributions to the platform.

The use of Zenodo makes it easier for everyone who has interest in the work of the project to see the concrete results of our research, but it is also an important tool to monitor and retrieve important analytical information that relates to scientific publications within the project. It is easy, for example, to evaluate the stream of publications and their trend throughout the project duration, or to retrieve information regarding paper impact and metrics.

Since January 2021 and in view of the final phase of the project, the Management Team has set up a project profile on Google Scholar, a platform commonly used by researchers to retrieve publication information about colleagues, but not widely used for institutions or single initiatives. The advantage we found in using this method is primarily the automatization in measuring key figures such as citations, which helps greatly in having a quantitative overview of the reach and impact of the single publication as well as of the project in general. Indicators like the h-index give a solid benchmark for evaluation and comparison with the rest of the community.

### 4.1.3 Twitter and website data

A significant part of the communication efforts of the APPLICATE team passes through the project website and its Twitter account. Both these spaces provide insightful analytics data that show the activity of the two platforms, its reception among the users and public, and the trend of this presence overtime. Indicators such as the engagement rate, the number of visits, geographical access and reactions to posts and campaigns can give a rather accurate description of how the project's channels of communication are working and how well they are serving the overall communication strategy.

While Twitter makes its analytics available through the embedded service "Twitter Analytics," each website can measure traffic and other data through various tools, which may come directly with the enabling platform like Wordpress, or via external providers. For APPLICATE, the chosen platforms to collect said data are Google Analytics and Matomo.

### 4.2 Stakeholder engagement

Stakeholder engagement activities are particularly important to assess the impact an initiative or a project has on the broader community and to understand the capacity of the project's team to interact across disciplines and topics. It is therefore key to find efficient ways to measure quantitatively and qualitatively the impact that these activities have.

### 4.2.1 Case Studies

The APPLICATE Case Studies are a series of self-explaining short documents covering a wide range of relevant topics in Arctic research and polar prediction, including the effects of Arctic environmental changes on mid-latitudes. APPLICATE develops case studies to show the use of weather, climate and sea ice information in the case of specific events with a significant impact on certain sectors or communities. The events analyzed in the case studies are selected together with users in User Group meetings, in thematic workshops, or through interviews.





The Stakeholder Engagement team in APPLICATE developed four case studies, a number that in itself can provide an indication of the level of involvement and influence that the team had with its users of reference. The impact of the case studies

has been assessed also by checking the amount of downloads through the website and the other distribution channels (e.g. Zenodo), the number of contributors that helped develop the studies, the number of channels in which they have been featured, the events and meetings in which these documents have been presented as well as how the information of the case studies have been used and applied in other contexts (see more about this under "Survey on Knowledge Transfer").

### 4.2.2 Blog Articles on Polar Prediction Matters

Polar Prediction Matters is a dialogue platform developed by the Polar Prediction Project that serves to enable exchanges between providers of polar weather and sea ice prediction products and the users of forecasts in the polar regions. As an important contributor in the field of polar weather and climate prediction and in an effort to enhance the reach of its stakeholder interactions, the APPLICATE team made several contributions to the platform, collaborating with users and presenting its own research dealing with applications of numerical predictions in the Arctic. Overall, the APPLICATE team made three

contributions to Polar Prediction Matters, and the impact of these has been assessed through the evaluation of analytics of the website and online interactions on social media and other APPLICATE spaces, as well as by evaluating the number of collaborators involved in the articles.

### 4.2.3 Policy Briefs

One of the objectives of European projects is to provide expertise and knowledge to inform policymaking bodies regarding

compelling issues and shape the future of research, directing science and policy towards new frontiers and relevant paths. While the relation between science and policy is not easy, many examples confirm the importance of scientific projects in shaping European and global policy (Lövbrand, 2011). In this perspective, the stakeholder engagement team has developed a series of policy briefs (Terrado et al., 2021), with the intention of summarising some of the key findings of APPLICATE's scientific efforts and translate them into accessible and useful information that could direct and inform policymakers. To assess how

impactful these documents will be, a look at the citations, especially in institutional documents and papers aimed at informing and shaping the future of research programmes, is fundamental, as well as an analysis of access data from analytics provided by website traffic and downloads. It is important to state here that, due to the publication of these briefs being so close to that of this paper, a full and comprehensive analysis of the documents' impact is still to be made.

### 4.2.4 Survey User Group

Members of the APPLICATE User Group (UG) participated in a survey that aimed to collect their feedback and assess their perception of how they have benefited from being part of the APPLICATE UG and how the interaction with them has been conducted during the past four years (Fig. 2). UG members indicated that they benefitted from participating in the project in





a variety of ways, including being regularly updated on the latest research in Arctic climate, learning about climate change issues and challenges faced by others in the Arctic while also widening their professional networks.

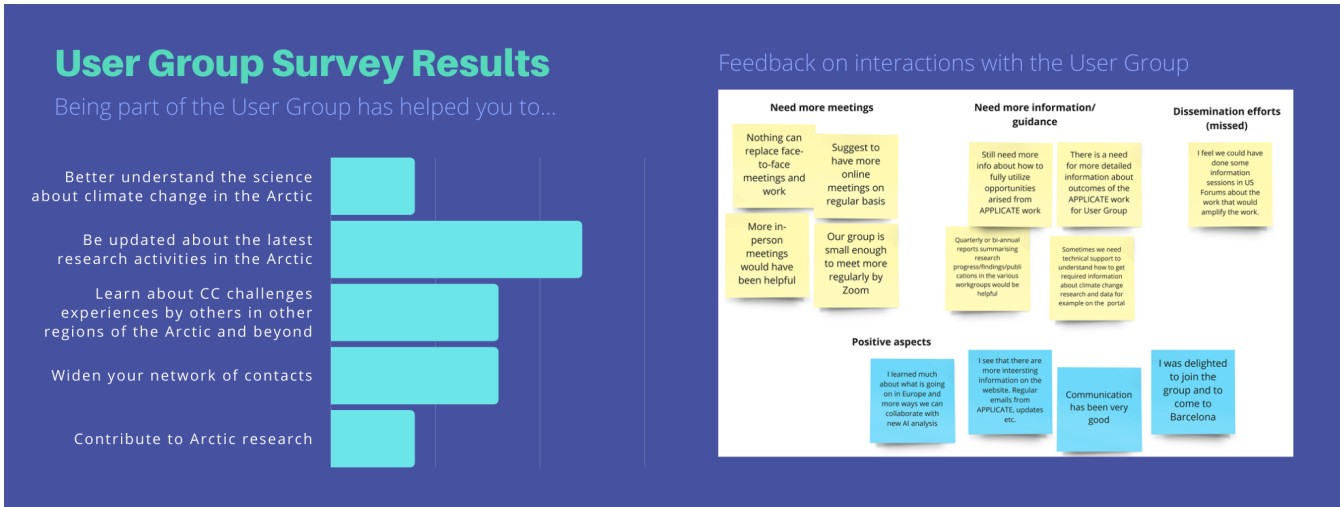

**Figure 2.** Main benefits of participating in the APPLICATE UG indicated by UG members. Sources of data and feedback has been collected and visualised by the team at the Barcelona Supercomputing Center (BSC-CNS).

## 4.3 Training

Educating the next generation of polar researchers has been one of the core aspects of the dissemination efforts within AP-PLICATE. The transfer of knowledge from mid-level and senior scientists to early career researchers has great potential for transmitting and spreading not only scientific results, but also research approaches and methodologies that have been developed within the project beyond the framework of the initiative. When measuring the reach of a project, its efforts in organising training events are important aspects to evaluate and assess.

The APPLICATE Education Team has organised three main activities:

1. the APPLICATE Webinar Series, a video collection outlining three of the main topics of the APPLICATE project;

2. the Polar Prediction School, in 2018 (in collaboration with the Polar Prediction Project and the Association of Polar Early Career Scientists APECS) (Tummon et al., 2018);

3. the APPLICATE-YOPP-APECS Online Course, held in 2019 to provide an overview of the state-of-the-art knowledge of Northern high-latitude weather and climate predictions.

By products resulting from some of these initiatives are video recordings (called "FrostBytes") illustrating research topics and questions on which early career researchers are focusing on, which are also used to determine the influence of the project on the community at large. Ways to evaluate training activities include, among others, a quantitative assessment of the students



attending these initiatives, which gives a good understanding of the breadth of the community that could be impacted, and the

monitoring of online performances of videos and materials produced during these activities. From a qualitative approach, the

APPLICATE team has used feedback surveys to evaluate the quality of the initiatives along with the effects that teaching and

training had on the research of the students, the likelihood of using the knowledge they gained in other applications etc. A

summary of these evaluations and of the impact of the training events in APPLICATE has been reported in one of the project's

deliverables (Schneider and Fugmann, 2020).

### 4.4 Clustering

Evaluating the results of a project's clustering engagements can provide a useful insight into how the project was able to

establish connections beyond its consortium boundaries and, therefore, can draw a picture of the spread of the scientific results

and methodologies of the project.

APPLICATE researchers have been involved in collaborations and cooperation with different actors and initiatives on mul-

tiple levels. The partners actively participating in the project have developed national links with key stakeholders or institutes

contributing to APPLICATE's research interests. The Clustering Team was particularly involved in engaging and strengthen

collaborations with other H2020 projects in Europe, participating in events and initiating clustering opportunities like the EU

Polar Cluster and the EU Modelling Cluster. Several scientific collaborations were supported by APPLICATE research, in

particular the Polar Amplification Model Intercomparison Project (PAMIP) (Smith et al., 2019) and the Sea Ice Model Inter-

comparison Project (SIMIP)(Notz et al., 2016). Strong ties have also been established with partners in North America and with

the endorsing Polar Prediction Project, the framework of YOPP (Jung et al., 2016).

All these initiatives have the potential to create not only a stronger modelling and polar community, but also a set of concrete

outcomes that can be presented as producing project impact. Among others, we measured the number of publications resulting

from the interaction with external partners (i.e., institutes and universities that were not receiving funds directly from APPLI-

CATE grant agreement), the amount of events and workshops organised with fellow projects thanks to the communication and

dissemination spreadsheet outlined in previous paragraphs, with an estimation of the amount of people reached by the event.

### 4.5 Survey on Knowledge Transfer

In order to implement a holistic evaluation of APPLICATE's efforts in knowledge transfer, in particular to acquire insights

into qualitative aspects of the activities that could not be represented by analytics and quantitative figures, in May 2021 the

team distributed an evaluation survey to APPLICATE's users and stakeholders contacts. These included some established

relationships (i.e., the User Group that supported APPLICATE in its stakeholder interactions) as well as external projects and

communities with which the project only partially interacted. The survey, shared through APPLICATE's online channels, was

meant to assess the knowledge of the community regarding materials and information disseminated through APPLICATE, and

understand the breadth of applications of the knowledge generated through the project in other initiatives, studies, policies

etc. The questionnaire was created using Google Forms and was adapted to fit the knowledge of the respondents. Alternating

Yes/No questions with multiple choice and open questions, the respondents could give feedback based on their knowledge




of the projects' dissemination efforts. The first question aimed to discern the respondents who were already familiar with APPLICATE and its information from those who did not know about the project's dissemination channels. The second question

asked specifically which of the KT materials (research publications, case studies, social media accounts etc) the respondents knew and if they had the chance to apply the information gathered from APPLICATE's KT activities in any other way. The third part of the survey assessed the main applications from external partners of APPLICATE's information, applications that went from research, to policymaking, to operational purposes and studies. One final question asked to briefly explain how the information acquired from APPLICATE was applied in any of the above-mentioned fields.

In total, 63 people took part in the survey. The results of this questionnaire highlight that the majority of the respondents did not know about APPLICATE's KT efforts and that the KT activities with which people were most familiar are those involving the project's researchers, i.e. research publications and presentations. Moreover, according to the responses, the knowledge disseminated through APPLICATE's KT channels has been applied mostly to other research efforts, followed by business applications, as graphically summarized in Fig. 3.

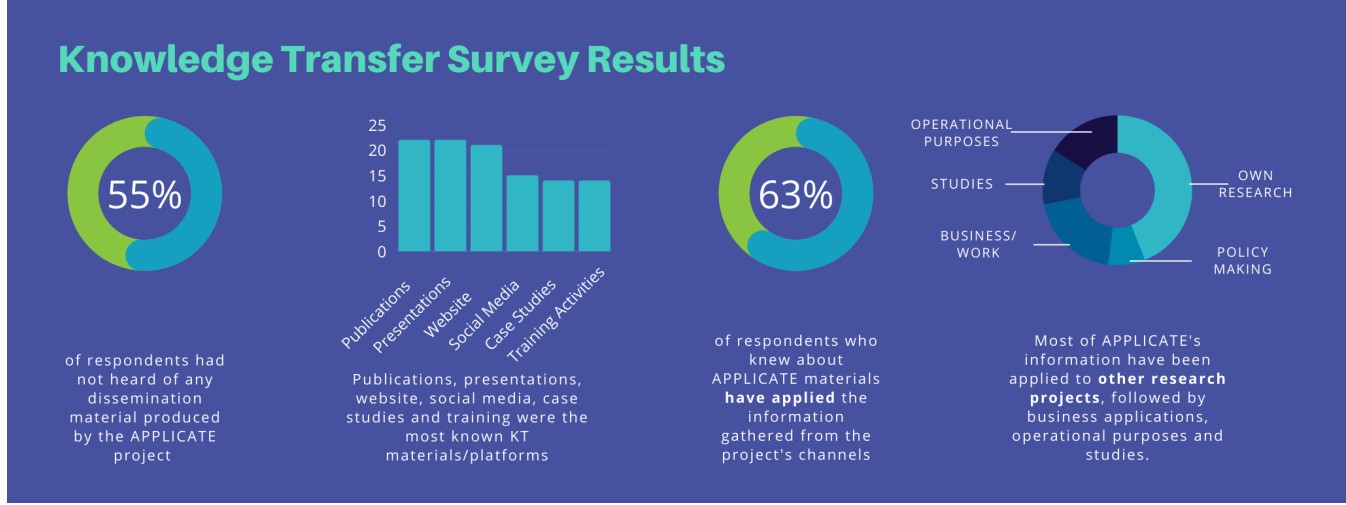

**Figure 3.** Summary of the main results of the APPLICATE's survey on Knowledge Transfer activities

When asked about the purposes of the applications of APPLICATE's information, most of the answers referred to the implementation of data or results from the project to answer different research questions, to guide operational developments and needs, to inform and contribute to the future research programmes. Moreover, many respondents declared to have used APPLICATE's KT materials in their own outreach efforts and to interact with policy. The word cloud in Fig. 4 was generated using the responses of the survey highlights the main applications of APPLICATE knowledge.





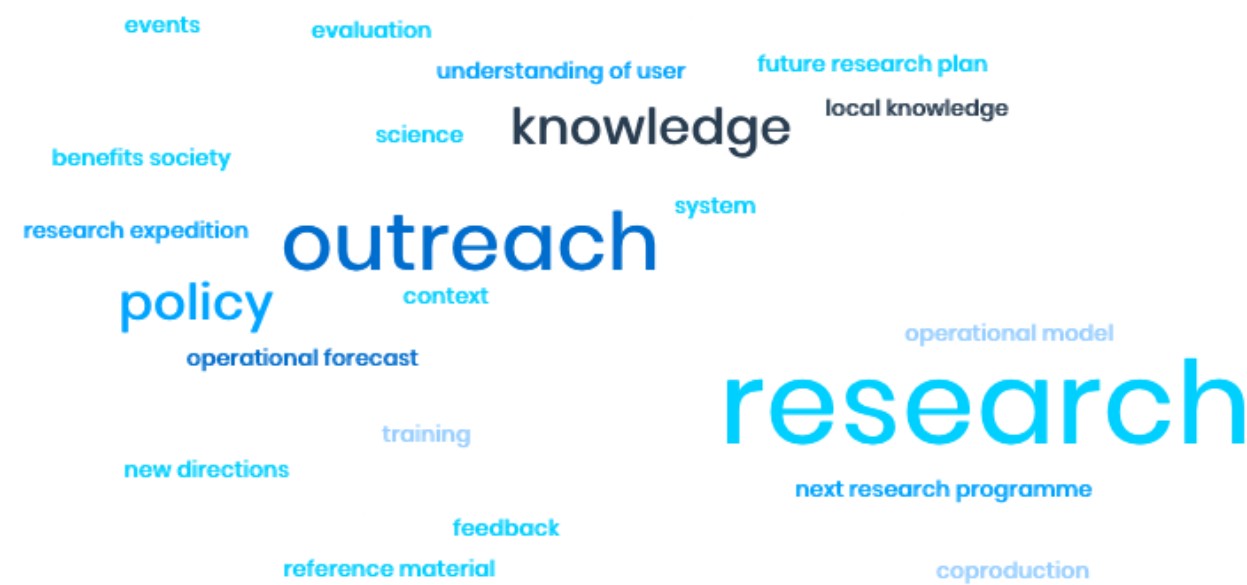

**Figure 4.** Word cloud featuring the most prominent themes in the answers related to the application of APPLICATE's knowledge

## 5 Monitoring and reporting impact for the APPLICATE project


Monitoring the trends and performance of online platforms and activities, documents, events and collaborations is helpful not only to assess the ultimate results of these activities, but it is fundamental to evaluate if the tools and strategies used for knowledge transfer are adequate and fit with the overall purpose of the dissemination plan and the project itself. It is helpful to establish an internal calendar to control the quantitative analytics and therefore assess the numerical performance of the

activities. In the case of APPLICATE, the project has been asked to complete three periodic reports, which were sent to the project officer at the European Commission as well as to two external reviewers. The reporting periods, which happened in intervals of 18 months from one another, provided a good occasion to run a general check on KT efforts, looking at e.g., the evolution in the number of followers on Twitter, the new events and documents developed since the last report, the results achieved compared to the plan and the grant agreement. Additional occasion for review have also been presentations and inter-

nal evaluation documents where the project and its work had to be outlined, happening once or twice a year. The benchmarks used to compare the performances were the measurements from previous recordings and figures from the project itself, given that the dissemination plan did not include either quantitative, or qualitative objectives against which analysing the activities. As it will be seen, this has been a setback in accurately measuring the impact of KT within APPLICATE.




## 6 Discussion

APPLICATE's methodology to develop and monitor its KT strategy was based on common practices implemented by the H2020 community as well as on requirements and recommendations coming from the funding agency, the European Commission (European Commission, 2015). This paper has been developed based on the experience of one single project, following the necessity to set a reference for future discussions and practices related to knowledge transfer and scientific research. In this section, we will present what have been the strongest and weakest points in our implementation of the methodology described in

the previous paragraphs, the lessons learned from the project and some recommendations for other project and communication managers dealing with KT in their projects.

### 6.1 Clear goals go a long way

Each project, however similar to others, has its own, unique set of objectives and purposes. Similarly, impact can be measured differently according to the project's ultimate goals. For this purpose, it is recommendable to start developing an impact strategy

at the beginning of the project outlining clear definitions for what it is meant by and how to measure impact, setting clear impact indicators. As clearly put in Jensen (2014): "Good impact evaluation requires upstream planning and clear objectives from practitioners", and the outcomes of this evaluation "should inform science communication practice." One risk when developing and using an impact tracking strategy is to end up with a set of quantitative and qualitative information that may mean little by themselves. Are 400 followers on Twitter half-way into the project a good number, or are two case studies too

little? Measurements and assessments acquire more meaning when related to a standard. This standard can be an external factor, for example common practices and measures from the funding agencies or a comparison with similarly organised projects, or it could be an internal scale, such as objectives outlined in the project's dissemination plan. When developing an impact plan for a research project, it is important to include some practical objectives to work as a benchmark to evaluate the performance of the initiative's KT efforts, as outlined in Fig. 5. This way, quantitative analytics assume a clearer value: to expand on the

Twitter example, 400 followers might indicate a successful strategy if the plan expected to reach ±50 profiles, but it might indicate a less strong impact if the team foresaw to approach 1000 individuals on that platform. For APPLICATE, a set of clear objectives was not set up in the first draft of the Dissemination Plan, thus making it harder to evaluate the development of the KT strategies throughout the project.

It is also important to underline, as mentioned, that some aspects of a KT strategy cannot be evaluated in a quantitative

manner and might, therefore, be less obvious and more challenging to measure. Depending on the project's idea of impact and impact objectives, each KT activity should actively contribute to the overall dissemination impact and serve a specific purpose, therefore describing how these actions fit into the overall strategy, to which target they aim and what scenario would be considered an impactful result is key to monitor and evaluate KT efforts.




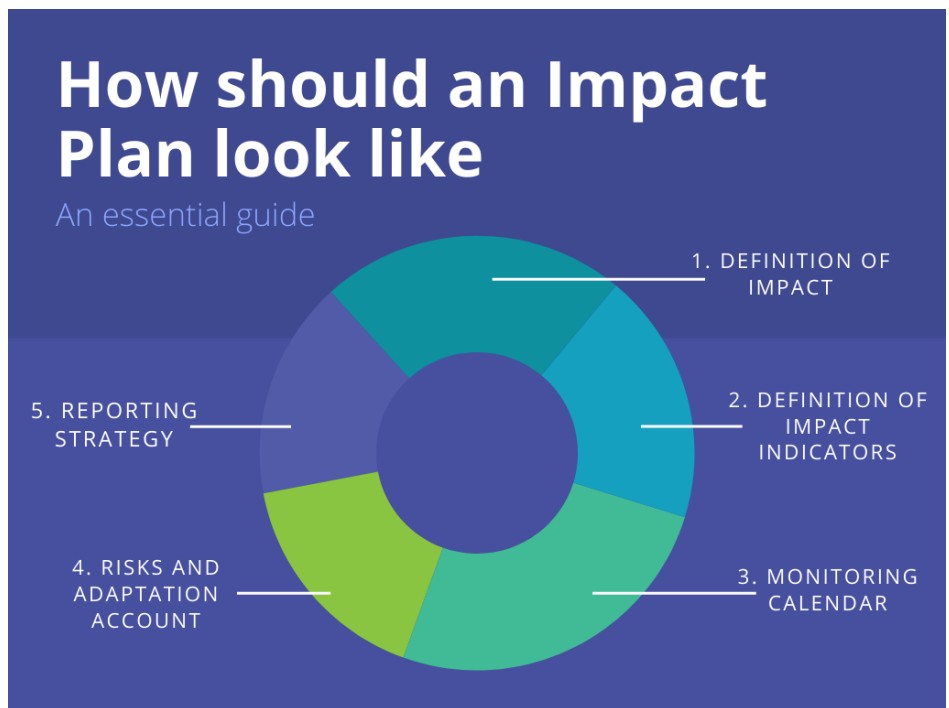

**Figure 5.** Graphic illustrating the five essential elements of an impact assessment plan

## 6.2 Monitoring as first step to reaching the goal

The natural structure of any European project, such as APPLICATE, requires periodic assessments on the progresses made by the team, including dissemination activities. This structure, in combinaton with an agile project management approach (Cristini and Walter, 2019), was instrumental to follow on the development and performance of the KT activities in APPLICATE with regularity and constance, to foresee possible issues and timely act on them. In the case of online outreach campaigns, monitoring the development on a regular basis allowed the team to make internal evaluations on the individual activities,

highlighting patterns (when and to whom is it best to address the campaigns?) and correcting practices. This is crucial to be able to change or adapt the strategy to meet the objectives of the communication plan and avoid reaching the end of reporting periods or the project itself with unaccomplished goals.

## 6.3 Involving researchers

Most of these tracking methods described in this paper require the involvement of project partners, and leaving the planning

and implementation of an impact strategy to the management or dissemination team may be not only an organizational burden, but also a strategic misstep. In the experience of the APPLICATE project, the involvement of project partners into tracking and evaluating KT activities has been fundamental. Particularly with respects to scientific publications and conference proceedings,





the active participation of each scientist into maintaining the internal tracking instruments such as the project's Zenodo archive and the table of dissemination and outreach activities allowed for a comprehensive and exhaustive assessment of the project's
dissemination reach. Involving each project partner in all the stages of the development of an impact plan is key to elaborate a strategic plan truly tailored for the needs and the capacities of a project.

## 6.4   The best comparison is with oneself

When developing an internal toolbox for the assessment of APPLICATE KT's impact, the project's team considered comparing some chosen indicators like the number of publications and citations and the size of the social media following with those of
similarly structured projects. While a project-to-project comparison could provide a seemingly objective benchmark to evaluate one's performance, we argue this is still a partial assessment of a project's impact. As mentioned, the impact evaluation strategy of each project can (and should) take into account different aspects, which might differ greatly between two initiatives even if they are conceived around similar foundations, topics or purposes. Two projects working to understand similar topics might want to address their impact in different ways, e.g. one could be interested in expanding the community of researchers dealing
with the topic and direct its KT towards clustering a new network of research while the other could invest in influencing the development of new policy and research programmes. For this reason, we also recommend developing internal benchmarks of impact assessment and compare the performance of KT activities to these values.

## 7   Conclusions and recommendations

In this paper, we summarised the efforts carried out within the APPLICATE project to disseminate and transfer knowledge on
weather and climate modelling in the Arctic and mid-latitudes along with the strategy developed by the project management and communication team to measure the impact of its KT efforts during the project's lifetime. Developing an impact plan for communication, dissemination, stakeholder engagement, training and clustering activities is a required, important step in the reporting process of a research project; but as we argue in this paper, it is an essential instrument to frame the whole KT strategy of an initiative, helps to set clear objectives and give direction to the actions undertaken in this framework.
The methodology presented here, albeit with some limitations as highlighted in the previous discussion, has proven itself very helpful for keeping track of the team's efforts in KT, monitoring the evolution of APPLICATE's activities towards the fulfilment of the overall goals and being able to respond and adjust the strategy when it was not the case. In particular, the instruments and approaches described here were developed following the project's needs and capacities, and were adequate in providing the required insights to the funding agency during the reporting phases. What the experience with the APPLICATE project
taught us is, above everything else, the importance of defining clear goals and indicators for impact assessment during the initial phase of developing a project's KT strategy, setting monitoring and reporting timeframes and involving the consortium in KT endeavours and tracking.

Although we argue in the previous paragraph against comparing the performance of different projects when it comes to KT results and activities, future actions from the funding agencies should include the development of a common set of criteria to



help project officers and managers to set up an adequate impact assessment plan for their projects, that reflects adequately not only the ambitions of their members, but also the capacities of the team and the objectives of the research programme.

## 8 Data availability

Sources describing the dissemination and communication efforts in the APPLICATE project can be retrieved on the project's page on the online repository Zenodo or go to the project's website: https://applicate-h2020.eu/.

*Author contributions.* SP summarised the introductory section, the Impact Tracking methods in Section 4, the discussion and conclusions. SP produced the graphics presented in this paper (Figure 2's data and data visualisation have been provided by Dragana Bojovic and Marta Terrado (Barcelona Supercomputing Center), and only adapted to the layout). LC led the writing of Sections 2 and 3. TJ proofread the whole paper.

*Competing interests.* The authors declare no competing interests.

*Acknowledgements.* The work described in this paper has received funding from the European Union's Horizon 2020 research and innovation programme under grant agreement No 727862. The authors acknowledge the precious contributions of Dragana Bojovic, Marta Terrado (both Barcelona Supercomputing Center) and Kirstin Werner (Alfred Wegener Institute Helmholtz Centre for Polar and Marine Research), for their friendly reviews during the preparation phase as well as the useful suggestions in making this work better.





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
