# Peer review of "How to get your message across? Designing an impactful knowledge transfer plan in a European project"

_Geoscience Communication, 2021_

## Author Comment (AC2)

**APPLICATE's Organization**

**WP1 WEATHER AND CLIMATE MODEL EVALUATION**

14 contributing institutions
28 team members

**WP2 ENHANCED WEATHER AND CLIMATE MODELS**

8 contributing institutions
32 team members

**WP3 ATMOSPHERIC AND OCEANIC LINKAGES**

11 contributing institutions
30 team members

**WP4 SUPPORT FOR ARCTIC OBSERVING SYSTEM DESIGN**

8 contributing institutions
17 team members

**WP5 IMPROVED PREDICTIVE CAPACITY**

7 contributing institutions
25 team members

**WP6 DATA AND HPC MANAGEMENT**

3 contributing institutions
11 team members

**WP7 USER ENGAGEMENT DISSEMINATION, TRAINING**

4 contributing institutions
9 team members

**WP8 CLUSTERING**

2 contributing institutions
17 team members

**WP9 PROJECT COORDINATION AND MANAGEMENT**

1 contributing institution
5 team members

---

## Author Response (AR1)

**gc-2021-29**    Submitted on 09 Sep 2021
How to get your message through? Designing an Impactful Knowledge transfer plan in a European Project
Sara Pasqualetto, Luisa Cristini, and Thomas Jung

**Reply on RC1**

Thank you for taking the time to read our paper and contribute to making it a useful contribution to the community. We will address the comments individually and provide our ideas for how to implement them in the text.

*The authors describe that APPLICATE was a collaboration among fifteen research institutions, universities, and national weather centers from eight European countries and Russia. The work performed with regards to knowledge transfer was primarily performed within work package 7. For a better understanding of the relative weight that was given to project management/knowledge transfer activities compared to the overall size of the project, it might be useful to see a diagram of the structure of the project and get some insight into the size of the team working on WP7. I believe this would be of interest, in particular, going forward with developing common best practices for future projects as well as giving the funding agencies insight into whether the size of the team and expertise within WP7 was sufficient or should be expanded.*

This is a valid point that was raised by the other referee as well. Adding more information on the project structure will help contextualize the project's breadth and relative effort from the various WPs, and an analysis into whether the project could have benefited from a bigger (or smaller) team will be added.
A paragraph to introduce the project's structure will be added:
"The project was researching Arctic transformations and their impact on lower latitudes by approaching the issues from different sides: nine work packages were established to answer and find solutions to critical questions related to, among others, model evaluation and development, predictive capacity, the Arctic observing system etc. The graphic below illustrates the topics and composition of the various work packages. *(see figure attached)* It is important to point out, however, that for work packages 7 and 8, although the coordinating aspects were curated by the members and institutions illustrated in the graphic, every person participating in any form to the project was asked to contribute to the KT and clustering efforts of APPLICATE."

*Related to the topic of team composition, it might be interesting to hear more details on what kind of expertise/background team members had and whether there is a need for not only expanding the team size and/or expertise of team members (e.g., social science, statistics, web metrics, search engine optimization for the website) but also developing more training opportunities for future project managers to fulfill the demands of outreach/communication activities on a project.*

Thank you for this suggestion. Text will be added to the paragraph introducing APPLICATE's knowledge transfer activities to present the team's expertise:
"These activities were carried out within the Work Package 7 (WP7), although each work package was required to contribute to the efforts of disseminating results and engage in outreach and education efforts. The team of WP7 included both social and natural scientists,

with expertise in communication and outreach strategies, co-production of climate services and training in research."

*In my opinion, it would also be of interest to elaborate more on how much of the activities within WP7 were self-directed by the team vs. prescribed by the funding agencies. The authors describe how important it is for the projects to develop plans to address the four focus areas of KT and ideally, funding agencies can give clear direction and support from the start to help with formulating plans and how to implement best practices that lead to project success. I believe it will be crucial as more and more large-scale scientific projects are created for the funding agencies to sufficiently support project management activities and also give more guidance and support for impact assessments and defining project success.*

Thanks for highlighting this point, which is very crucial for our analysis. The relation with the funding agencies, especially in the context of European projects, is fundamental and should be pivotal in developing a successful strategy of impact assessment. We briefly mention how some of the activities and tracking methods have been suggested by the funding agency, but we can expand on where the support could be stronger and which aspect would need more structure.

**Reply on RC2**

Thank you for the thorough review of our paper and for the useful suggestions made. We will address the comments individually and provide our ideas for how to implement them in the text.

*I encourage authors to re-read the MS and consider shortening / streamlining.*

*Title: The title specifically mentions EU projects. Is there anything that can be done to make the findings relevant for non-Eu funded projects?*

The title mentions EU-projects specifically because the project used as reference, i.e. APPLICATE, is a European project and specifically experienced the dynamics and structures that are typical of a EU-funded project. While many aspects can be applied to other contexts and be "universalized" to research-projects in general, we thought it useful to specify the kind of context that we are navigating as some of the structures and institutions we mention refer specifically to the EU-funded experience.

*Abstract: If space allows, considering adding 1-2 sentences at the end of the abstract text summarizing key messages and findings to make the abstract more focused and useful.*

Thank you for suggesting this. We plan to add a summarizing sentence at the end of the abstract:

"Our experience found that an assessment strategy should be included in the planning of the project as a key framing step, that the individual project's goals and objectives should drive the definition and assessment of impact, and that the researchers involved are crucial to implement a project's outreach strategy."

*Line 44 – Remove double parenthesis*
The correction will appear in the final version of the manuscript.

*Line 48 – Should it be "Indicators" rather than "Indications"?*
It should indeed be indicators, it will be implemented in the final version of the manuscript.

*Line 49 – "Roadmap" instead of "Road map"?*
The correction will be implemented in the final version of the manuscript.

*Line 57 - As a person not familiar with the project structure, I feel like it's time to introduce the structure oof the project at this stage, incl. a figure/graphic summarizing linkages between different WPs, User Group, and teams (KT, Communication, User Engagement, Education, Clustering, etc. )*
This point was raised by the other referee as well, and it is a very valid one. We will add a figure illustrating the project structure (see attached).

*Line 63 – Stakeholders and users are mentioned in this paragraph, but the definition and clarification comes much later (Lines 144-145). Consider defining both already here.*
Thanks for the suggestion, we will move up the definition for clarity.

*Line 104 – out of curiosity, was there a coordination between different WPs when it comes to dissemination of results and engagement with community?*
The team within WP7 was responsible for planning and coordinating dissemination and outreach efforts directed to all target groups and were in charge of involving the other WPs and their researchers in these efforts.

*Line 108 – could please provide several examples of "targeted activities"?*
We are adding a sentence to elaborate on the term "targeted activities": "Targeted activities included dedicated meetings and workshops with users to update on the progress and results of the project, policy events to engage with policymakers within EU institutions, press releases to appear in news outlets addressing non-scientific public."

*Line 108 – Is the word "team" missing after the word "... dissemination"?*
Yes, the correction will appear in the final version of the manuscript.

*Line 135 – how were policy briefs communicated?*
Policy briefs have been introduced during policy meetings organized by the EU and the project as well as during workshops like the PPP-SERA workshop in the framework of the Polar Prediction Project. The documents are still available on the project's website for download. We added a clarification on the text as well that will appear in the final version of the manuscript.

*Line 146 – how do you define "pro-active"?*
It is linked with the co-production approach that has been applied by the stakeholder engagement team, and refers to the active engagement of users as stakeholders in the evaluation and development of climate services for example, asked to give feedback and contribute to the development of climate products and services that respond to the actual needs of the community.

*Line 160 – NGOs and public sector are not mentioned among UG members. Was it the case? If so, was it intentional? I am just curious here.*

The idea was for the UG to be a small team to facilitate discussion and interaction with the coordination team and the stakeholder engagement team and so a limited number of members were invited based on their background, expertise and role in the Arctic community. The public sector (local community managers) and few NGO's (environmental conservation) have been approached and invited to join the UG but invitations have either been declined or not followed on as they did not bring the level of engagement required for the project.

*Line 161 – word "external" is used twice*
We will implement the correction in the final version of the manuscript.

*Line 165 – please give 1-2 examples here?*
For example, an exercise with the UG involved defining which environmental parameters (e.g., sea-ice extent, number of frost days) would be useful for them to know in advance to plan business operations (e.g., shipping, reindeer herding). It will be added in the final version.

*Line 167 – either "data" or "Information"?*
Yes, the correction will appear in the final version of the manuscript.

*Line 196 – even though readers can check Schneider & Fugmann (2020), I encourage authors to add 1-2 sentences key outcomes of the survey.*
We added the following sentence:

"Among the end results of the courses, it is interesting to highlight that in every case, students and participants really appreciated the syllabus and content of the trainings, in particular the combination of theoretical and practical aspects, while lessons learned were mostly of logistical nature, such as improvements in the organization of the event."

*Line 214 – consider adding 1-2 examples of clustering activities here or earlier in this section.*
Clustering activities in this case were related to the cooperation with other projects from the EU polar cluster, for example participating in each other's project meetings, by organizing outreach and policy events with them and by consolidating cooperation with other projects beyond Europe by coordinating papers and initiatives in the framework of the Polar Prediction Project, among others. We listed them in this paragraph, but we will consider expanding and add more specific examples.

*Lines 217-220 – this part highlights the challenge of impact tracking during the timeline of the project. The definition of impact in this work (and project) is different compared to the one mentioned in Lines 22-23. With the definition of impact by the EU commissions, I wonder, if the impact can be fully assessed during the project timeline. Can authors think of a recommendation for the funding agencies here?*
This is a discussion that has been going on with the funding agencies also during the final review. What we focus on this paper are outreach activities and their evolution during the project timeline, and this is something we believe is possible to assess, with the help of a common strategy and approach. We did not expand on all those impacts that go beyond the final date of the project, simply because, with the project finishing relatively recently, these are still largely unforeseen. However, it is a point that needs attention and is one of the most critical aspects that have been debated, and for which a clearer guidance from the funding

agencies is needed. We will consider adding this as a point/recommendation directed to funding agencies.

*Lines 223-319 – the sub-sections are defined based on the type of activity. I wonder whether authors considered arranging this part of the MS based on the target audience?*
The organization of sections into type of activities was made because other categorizations (for example, by target audience) would involve a repetition of descriptions as some of these activities are applied to different audiences. Moreover, when illustrating the methodologies to assess the impact of the various activities, it was easier to refer to them per type of outreach effort rather than to whom the activities were addressed.

*Lines 231-232 – how does one monitor and evaluate meetings and workshops?*
We monitored meetings by logging all the events, conferences and workshops to which project members took part or organized in a database, the one illustrated under paragraph 4.1.1. One way to evaluate them, and this applies only to the events that were organized by the project, is to administer questionnaires asking for feedback and impressions on the organization, content and impact of the meeting. This last point, as we described in the paper, was not always applied in the context of APPLICATE, but would have been an important source of qualitative information on the impact of the events.

*Line 234 – events and initiatives*
The correction will appear in the final version of the manuscript.

*Line 236 – what are the criteria required by the European commission?*
In the portal set up by the EU to which each project has access and where projects share documents and information, when submitting periodic reports projects are asked to specify details related to events organized and attended by project's members, and include the type of outreach activity, title and logistical details such as where and when the event took place, and the target audience. The criteria are listed in the paragraph.

*Line 245 – Items #5 is not reflected in the Figure 1. Is it on purpose?*
We thought of grouping this information under the audience details, we will make an addition to the figure to specify this point.

*Line 283 – How short?*
The longest case study is 9 pages long – we added the information to the text.

*Lines 312-313 – Again, authors highlight the challenge of timeline. Do you have any recommendations or solutions to solve the issue?*
See comment to lines 217-220

*Line 315 – it would be interesting to see a breakdown of UG members per sector.*
We decided not to focus too much on the description of the user groups and stakeholders as we thought it would not have added too much in evaluating and analyzing our strategy of impact assessment, but we will consider specifying more clearly when mentioning our User Group which sectors were they coming from.

*Line 332 – byproducts?*
The correction will appear in the final version of the manuscript.

*Line 340 – consider adding 2-3 sentences summarizing takeaway messages from Schneider & Fugmann (2020)*
We plan to add a short sentence to recall a previous explanation of the results: "The results of these surveys highlight, among other aspects, a general appreciation for the content tackled during the trainings and emphasized a need for more occasions to confront with senior scientists, in addition to evaluation of more organizational aspects."

*Lines 341-357 – I wonder what was the main driving force of clustering activities? Was it more of top-down approach or rather bottom-up, with cases when researchers in different EU (and non-EU projects) found ways for synergies and cooperation irrespective of planned clustering activities? Could you please comment on that?*
Clustering activities targeted specifically projects and initiatives with US and Canadian institutions in the framework of a strategy implemented and wanted by the EU and exposed in the call for proposals.

*Line 350 – a space missing before (Notz et al ...)*
The correction will appear in the final version of the manuscript.

*Line 358 – suggest introducing KT abbreviation here or earlier in text and use it throughout the MS*
We will implement the suggestion in the final version of the manuscript.

*Line 361 – is "APPLICATE's users and stakeholders contacts" the same as the UG?*
Yes it is, we will specify it in the text.

*Lines 375-380 – I must admit that I am surprised that 55% of respondents had not heard of any dissemination material produced by the project! Are there any lessons one can learn from that? Any recommendations can be given?*
We asked ourselves this and we thought that there are many ways to interpret this. One could relate to the project's strategy not being as successful, or it might be due to the fact that we shared the survey through channels and groups which were not directly addressed by our outreach activities. Either way, we thought it was an evaluation that went beyond the illustration of an impact strategy, which is ultimately the goal of the paper, and that is why we did not include any explanation or interpretation for these figures.

*Figure 3 – please add Y axis on graph 2*
We will implement the suggestion in the final version of the manuscript.

*Figure 4 – the figure takes a lof of space. Consider removing it and rather mentioning 4-5 key words*
Thanks for this suggestion, we will consider removing the image which indeed takes a lot of space and could be explained with some lines of text.

*Lines 420-423 - I agree that absolute numbers are not always useful. Other indicators, like engagement rate or other similar values divided by the number of followers are more representative.*

*Line 434 – Evaluations of*
We will correct it in the final version of the manuscript.

*Lines 458-476 – Consider splitting this part into two (or several), for examples recommendations for funding agencies and recommendations for project managers.*
By splitting the section, it seems to us that the discourse would become too fragmented and the link between the recommendations stemming from the conclusions vaguer. However, we can add a sentence relative to recommendations for project and communication managers: "In addition, project and communication managers should try to develop best practices within their community, for example by sharing lessons learned in their projects through publication and presentation at conferences and events."